# Molecular dynamics simulations reveal the selectivity mechanism of structurally similar agonists to TLR7 and TLR8

Xiaoyu Wang[1], Yu Chen[1], Steven Zhang[2], Jinxia Nancy Deng[3]*

**1** Computational Chemistry Department, Shanghai ChemPartner Co., Ltd., Shanghai, China, **2** Chemistry Department, Shanghai ChemPartner Co., Ltd., Shanghai, China, **3** Computational Chemistry Department, ChemPartner, South San Francisco, CA, United States of America

* ndeng@chempartner.us

**Data Availability Statement:** All relevant data are within this paper and its Supporting information files. The trajectory files generated in the MD simulation are available from Zenodo DOI: 10. 5281/zenodo.5959193.

## Abstract

TLR7 and TLR8 are key members of the Toll-like receptor family, playing crucial roles in the signaling pathways of innate immunity, and thus become attractive therapeutic targets of many diseases including infections and cancer. Although TLR7 and TLR8 show a high degree of sequence homology, their biological response to small molecule binding is very different. Aiming to understand the mechanism of selective profiles of small molecule modulators against TLR7 and TLR8, we carried out molecular dynamic simulations on three imidazoquinoline derivatives bound to the receptors separately. They are Resiquimod (R), Hybrid-2 (H), and Gardiquimod (G), selective agonists of TLR7 and TLR8. Our MD trajectories indicated that in the complex of TLR7-R and TLR7-G, the two chains forming the TLR7 dimer tended to remain "open" conformation, while the rest systems maintained in the closed format. The agonists R, H, and G developed conformational deviation mainly on the aliphatic tail. Furthermore, we attempted to quantify the selectivity between TLR7 and TLR8 by binding free energies via MM-GBSA method. It showed that the three selected modulators were more favorable for TLR7 than TLR8, and the ranking from the strongest to the weakest was H, R and G, aligning well with experimental data. In the TLR7, the flexible and hydrophobic aliphatic side chain of H has stronger van der Waals interactions with V381 and F351 but only pick up interaction with one amino acid residue i.e. Y353 of TLR8. Unsurprisingly, the positively charged side chain of G has less favorable interaction with I585 of TLR7 and V573 of TLR8 explaining G is weak agonist of both TLR7 and TLR8. All three imidazoquinoline derivatives can form stable hydrogen bonds with D555 of TLR7 and the corresponding D543 of TLR8. In brief, the set of total 400ns MD studies sheds light on the potential selectivity mechanisms of agonists towards TLR7 and TLR8, indicating the van der Waals interaction as the driving force for the agonists binding, thus provides us insights for designing more potent and selective modulators to cooperate with the hydrophobic nature of the binding pocket.

**Funding:** Chempartner provided support in the form of salaries for Xiaoyu Wang, Yu Chen, Steven Zhang, Jinxia Nancy Deng, but did not have any additional role in the study design, data collection and analysis, decision to publish, or preparation of the manuscript.

**Competing interests:** Commercial affiliation does not alter our adherence to PLOS ONE policies on sharing data and materials. The authors declare no competing financial interest.

## 1. Introduction

Toll-like receptors (TLRs) are a large family of proteins, playing an important part in innate immune system, that recognize structurally conserved molecules, such as single-stranded (ss) or double-stranded (ds) RNAs or DNAs, lipoproteins and lipopolysaccharides derived from microbes and then activate immune cell responses [1]. A typical TLR is a single-spanning receptor consisting of three domains: an extracellular domain (ECD) with variable number of leucine-rich repeat sequences (LRRs) for the recognition of pathogen-associated molecular patterns (PAMPs), a transmembrane domain (TMD), and an intracellular Toll-interleukin 1 receptor (TIR) domain initiating downstream signaling [2]. Until now, thirteen TLRs have been identified, and among which, TLR3, TLR7, TLR8, and TLR9 are located in intracellular membranes because they are sensors of nucleic acids. Specifically, TLR3 recognizes viral dsRNA, and TLR9 senses unmethylated cytosine phosphate guanosine (CpG) containing DNA, whereas TLR7 and TLR8 both are located in endosomal membrane and function as viral ssRNA sensors [3].

Many studies have revealed that the expression levels of TLR7 and TLR8 are altered in some autoimmune diseases, such as arthritis, cancers [4–8], or in antiviral regimes, including corona virus prevention and HIV [9]. Thus, novel drug design and development against TLR7 or TLR8 became very attractive.

In the last few years, many TLR7 or TLR8 agonists with different scaffolds have been developed. These agonists leading to the induction of certain IFNs, cytokines and chemokines can be applied to the treatment of some diseases and can be used as good adjutants of vaccines [10–12].

The drug development of TLR modulators requires a solid understanding of TLR7 and TLR8 activity regulation. TLR7 and TLR8, sharing high degree of sequence homology and three dimensional structure similarity, are both known to serve as endosomal pattern recognition receptors (PRRs) for a number of RNA viruses, such as HIV, coronaviruses, influenza etc [9, 13–16]. However, there still exist many distributional and functional differences between these two closely related proteins. TLR7 is mainly expressed in plasmacytoid dendritic cells (pDC) and B cells [17–19]. But TLR8 is mainly expressed in myeloid dendritic cells (mDC), monocytes, macrophages and neutrophils [17, 20, 21]. TLR7 recognizes guanosine and its derivatives [22]while TLR8 serves as a uridine receptor [23]. Moreover, interferons induced by pDC are the major production of TLR7 signaling [24], whereas TLR8 signaling mainly results in NF-kB pathway activation and subsequent proinflammatory cytokines and chemokines expression [25]. To understand how TLR7 and TLR8 can recognize different ligands, and as consequence to activate different signaling pathways, we design computational simulations to investigate the selectivity mechanisms of small molecule agonists of TLR7 and TLR8.

Molecular dynamics (MD) simulation is a very established computational technique to understand the protein structure-function relationship and guide the drug design. MD simulation also has played an important role in characterizing receptor-ligand interaction [26–28], providing the guidance of structure-based drug design [29–31] and typical conformations for virtual screening [32–37]. Besides, it helps to reveal the novel binding sites which have not been captured by NMR and X-ray crystallographic analysis, for example, cryptic binding sites in HIV-1 integrase [38–41]. Up to date, MD simulations have been successfully applied to many large systems, such as the complete HIV1 capsid with 64 million atoms up to 100ns [42–45]. MD simulation has been also instrumental on understanding the protein folding and function regulation with a simulation time of 10-100us [46–48].

Because of the above mentioned success, not surprisingly, some MD simulations were carried out on TLRs, such as the stability of vaccine and TLRs [49, 50], TLRs model [51, 52] and

**Fig 1. The chemical structures of the three agonists: Resiquimod (R), Hybrid-2 (H) and Gardiquimod (G).**

effect of mutations on TLRs dimer [53]. MD simulation was applied to equilibrate homology model of TLR7, proposing the appropriate TLR7 dimer structure and studying the binding site and residues significant for dimerization [54]. And it was used to explain the difference of interaction mode between agonists and antagonists (including imidazoquinoline and adenine derivatives) for TLR7 [55]. Targeted molecular dynamics (TMD) was employed to study the conformational transition of TLR8 dimerization, illuminating the internal mechanism of relatively aggregate movement of two TLR8 chains [56]. These research results have inspired our research ideas.

Some imidazoquinoline derivatives (Fig 1), i.e. Resiquimod (R) [57, 58], Hybrid-2 (H) [59] and Gardiquimod (G) [60] are agonists of TLR7 and TLR8. They carry the common core of the same parent nucleus, whereas the side chain is oxygen, carbon, nitrogen atom, respectively. However, this one-atom difference in the chemical structure results in magnitude difference in their biological activity on TLR7 and TLR8. In general, they are more potent on TLR7 than TLR8 (Table 1). These two facts aroused our interest in investigating the agonist selectivity for TLR7 and TLR8 by MD simulations.

In this work, initially, eight systems were built, and they are TLR7 (apo), TLR7-R, TLR7-H, TLR7-G, TLR8 (apo), TLR8-R, TLR8-H, and TLR8-G. Thereafter, 50ns MD simulations of each of the eight systems were performed. Subsequently, the binding free energy of each ligand with TLR7, and TLR8 was calculated respectively using the MM-GBSA method. The interactions between the three agonists with TLR7 and TLR8 were analyzed to explain the intrinsic mechanism of the selectivity of the three agonists at the atomic level.

## 2. Materials and methods

### 2.1 Research systems

To investigate the selectivity of R, H and G with TLR7 and TLR8, respectively, the eight systems for the study are TLR7 (apo), TLR7-R, TLR7-H, TLR7-G, TLR8 (apo), TLR8-R, TLR8-H

**Table 1. Comparison of predicted binding free energies and experimental EC50 values of R, H and G with TLR7 and TLR8.**

| Compound | TLR7 | | TLR8 | |
|:---:|:---:|:---:|:---:|:---:|
| | Predicted binding free energy (kcal/mol) [a] | EC50 (nM) | Predicted binding free energy (kcal/mol) [a] | EC50 (nM) |
| **R** | -44.21 ± 0.24 | 1400 [78] | -39.55 ± 0.20 | 6400 [78] |
| **H** | -50.50 ± 0.25 | 2.5 [59] | -42.09 ± 0.28 | 19 [59] |
| **G** | -33.65 ± 0.31 | 2000 [79] | -20.99 ± 0.19 | No activation of NF-κB [80] |

[a] The predicted binding free energies were obtained based on 30–50 ns MD simulation trajectory. A total of 201 snapshots evenly extracted from the 30–50 ns MD trajectory of each complex system were used for MM-GBSA calculations.

and TLR8-G. The crystal structures obtained from the Protein Data Bank (PDB) were used as the templates, and the codes are 5GMH (TLR7-R) [22], 5ZSG (TLR7-G), 3W3N (TLR8-R) [61], 4R6A (TLR8-H). The apo TLR7 was modeled from the X-ray complex of TLR7-R (5GMH) by removing the ligand and same to the apo TLR8 from TLR8-R (3W3N). The TLR7-H system was modeled based on coordinates of TLR7-R by replacing the oxygen on aliphatic chain of R to carbon atom. Similarly, the TLR8-G system was modeled by replacing the oxygen atom on aliphatic chain of R in TLR8-R with a nitrogen atom. The crystal structures were connected for the missing residues via SWISS-MODEL server [62]. The modeled structures were aligned to their crystal structures and were retained with corresponding ligand. The sequence fragment between amino acid residue 434–458 was removed to align with the biological understanding that those motifs in both TLR7 and TLR8 are cleaved prior to the activation process [61, 63]. In fact, these residues are also missing in 3W3N and 4R6A crystal structures. With reference on the recent research that human TLR7 and TLR8 proteins are in acidic endolysosom (pH is around 5) [64, 65], the protonation states of agonists were predicted by Maestro (from Schrödinger V.2019-4) [66] at the pH value of 5 (±0.5). The nitrogen atoms on quinoline ring in all complex systems and nitrogen atoms on aliphatic chain in TLR7-G system and TLR8-G system are protonated.

## 2.2 Molecular dynamics simulations

All MD simulations were carried out in the isothermal isobaric (NPT) ensemble with periodic boundary condition by the program GROMACS (version 2020.4-GPU package) [67]. The AMBER ff99SB force filed [68] was applied to model proteins and the general amber force filed (GAFF) [69] was assigned to model the small molecule agonists. Each system was placed in a rectangular box of TIP3P explicit water molecules with a minimum distance to the water box wall of 9 Å [70], and counterions (Cl-) were added to neutralize the system. Each simulation system was first subjected to energy minimization using the steepest descent algorithm. Then, a simulation was carried out to heat the system to 300K with the protein fixed using a harmonic restraint. The temperature was kept close to 300K by V-rescale thermostat [71] and the pressure was kept at 1 bar using the Parrinello-Rahman pressure coupling scheme [72]. The LINCS method [73] was used to restrain bond lengths that including hydrogen atoms, allowing an integration step of 2 fs. Long-range electrostatic interaction was calculated using PME with the cutoff 9Å. Finally, based on the relaxed system, the simulation so called the production phase was performed without any constraints for 50ns.

## 2.3 Binding free energy calculation

The MM-GBSA binding free energy calculations between the protein and agonist were using MMPBSA.py module in AmberTools20 package. The binding free energy ($\Delta G_{binding}$) between a receptor and a ligand can be estimated using Eqs 1 and 2:

$$\Delta G_{binding} = \Delta E_{MM} + \Delta E_{solv} - T\Delta S \tag{1}$$

$$\Delta G_{binding} = \Delta E_{elec} + \Delta E_{vdW} + \Delta E_{polar} + \Delta E_{surf} - T\Delta S \tag{2}$$

$\Delta G_{binding}$: the relative binding free energy. $\Delta E_{MM}$: the gas phase energy consisting of electrostatic ($\Delta E_{elec}$) and van der Waals ($\Delta E_{vdW}$) terms. $\Delta E_{solv}$: solvation energy including both the polar solvation energy, $\Delta E_{polar}$ and the nonpolar solvation component, $\Delta E_{surf}$. The $\Delta E_{polar}$ in above equation is calculated by the GB model [74] and $\Delta E_{surf}$ is estimated by the solvent accessible surface area (SASA). $T\Delta S$: the entropy term. It is usually suitable for systems with large

conformational changes, thus was ignored in our simulations, aligning with other previous computational studies [75, 76].

## 3. Result and discussion

### 3.1 Characterization of conformational changes in the systems

The conformational changes of the eight simulated systems, ie TLR7 (apo), TLR7-R, TLR7-H, TLR7-G, TLR8 (apo), TLR8-R, TLR8-H, TLR8-G systems were first analyzed in terms of the root mean square deviation (RMSD) of protein backbone in each system (Fig 2). The RMSD values of TLR7 (Fig 2A) in the three complex systems change from the initial value (0 ns) to a scope of 2.5–4.0 Å, maintaining in this scope after 30 ns. Among these three, TLR7-G system, represented by the green line has the largest RMSD value close to 4 Å, and therefore the largest deviation comparing with the rest agonists. Throughout most of the simulation, TLR7-R and TLR7-H systems have similar RMSD plots to one another.

The RMSD values of TLR8 (Fig 2B) in the three complex systems change from the initial value (0 ns) to a scope of 2.0–3.0 Å and then also maintain in this scope after 30 ns. Among these three, TLR8-R, TLR8-H and TLR8-G systems have the similar RMSD plot during most of the simulation time. These reveal the larger conformational change of complex systems presented on TLR7 compared to that of TLR8. It can be observed that the RMSD value in the TLR8 (apo) system is slightly higher than those of the TLR8 complex systems after 30 ns, which indicates that the TLR8 is more flexible in the absence of agonist. The RMSD value in the TLR7 (apo) system is slightly lower than those of the TLR7 complex system after 30 ns, which indicates that the TLR7 is less flexible in the absence of agonist.

In addition, the trajectories displayed in Fig 3 using VMD software [77] to observe conformational changes show that the two chains (i.e. chain A and chain B) in TLR7 of the TLR7-R and TLR7-G systems were gradually open measured by the distances between the anchor points in the first 30 ns, but the TLR7-H system maintained the closed conformation state. In order to describe the closed and open conformational changes, the centroid of R784 backbone atoms in chain A and chain B of TLR7 were defined as anchor points to calculate the time-dependent distance between two chains (Fig 3A). In the same way, the centroid of P773, the corresponding counterpart in TLR8 of R784 in TLR7, backbone atoms in chain A and chain B of TLR8 were chosen as anchor points (Fig 3B). These two pairs of amino acid residues were

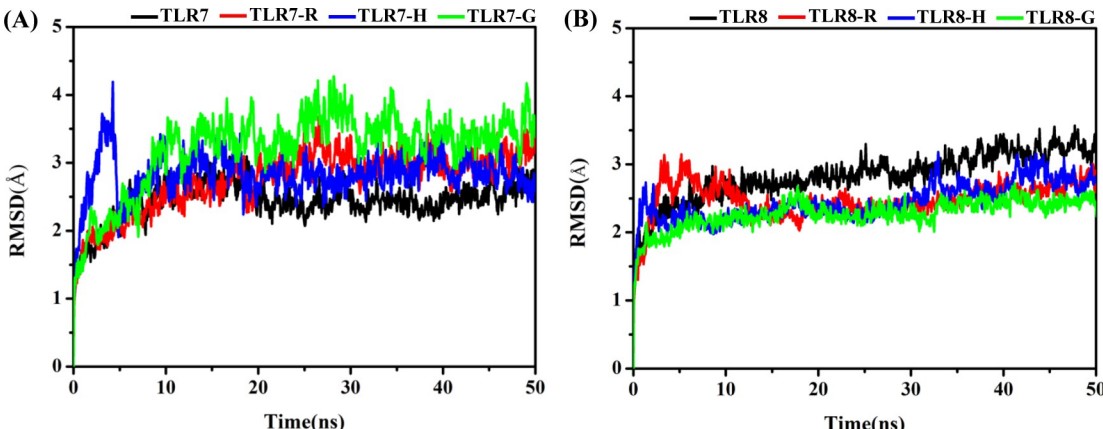

**Fig 2. The overall RMSD values of TLR7 and TLR8 with and without the agonists.** (A) TLR7 (apo) system, TLR7-R system, TLR7-H system and TLR7-G system. (B) TLR8 (apo) system, TLR8-R system, TLR8-H system and TLR8-G system.

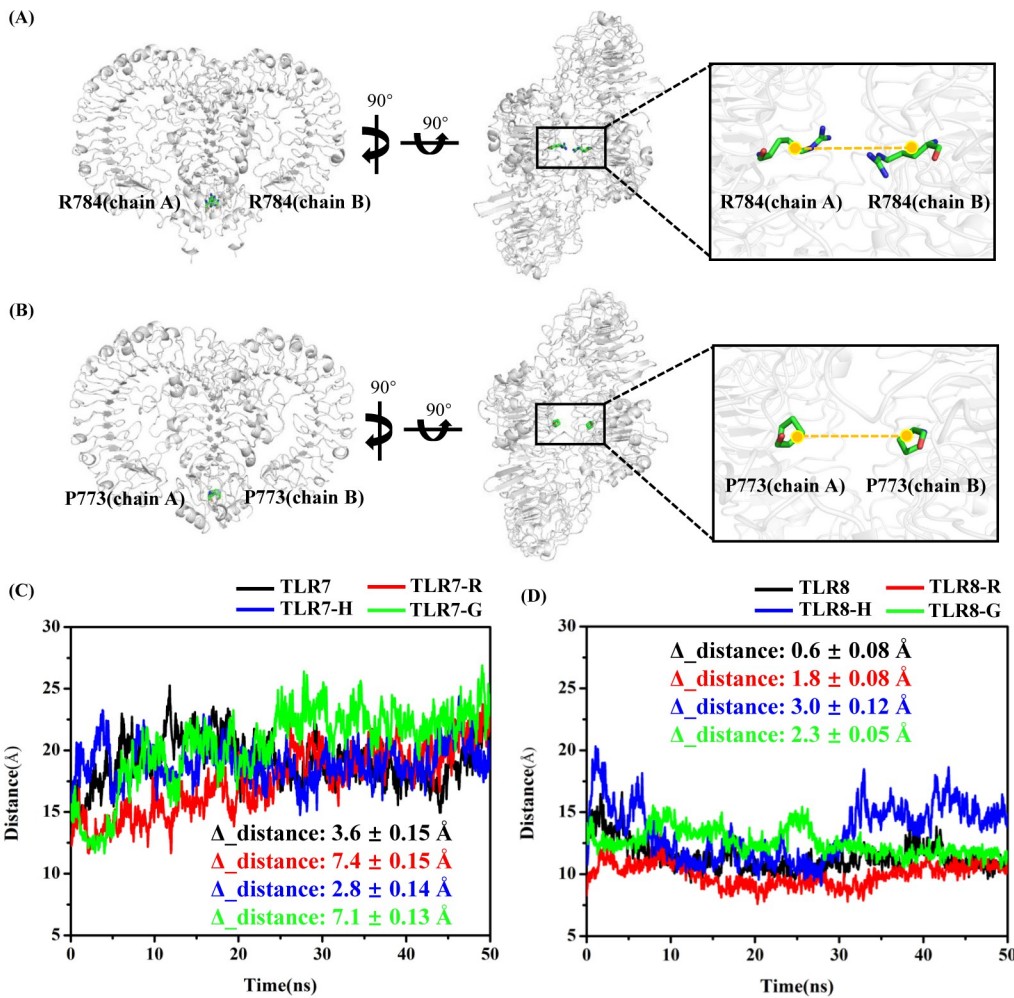

**Fig 3. The distance changes of chain A and chain B for TLR7 and TLR8.** (A) The R784 of each chain is defined as anchor point on TLR7. (B) The P773, counterpart of R784 of TLR7, of each chain is defined as anchor point on TLR8. (C) The time dependent distance between chainA-R784 and chainB-R784. (D) The time dependent distance between chainA-P773 and chainB-P773. The Δ_distance refers to the difference of the chain A and chain B distance between the structures during the last 20 ns MD simulations and initial structure. The initial distances (0 ns) are 15.1 Å, 12.4 Å, 16.5 Å and 15.3 Å in TLR7 (apo), TLR7-R, TLR7-H, TLR7-G systems, respectively. The initial distances (0ns) are 11.9 Å, 8.3 Å, 11.9 Å and 14.1 Å in TLR8 (apo), TLR8-R, TLR8-H, TLR8-G systems, respectively. The average distances during the last 20 ns are 18.7 ± 0.15 Å, 19.8 ± 0.15 Å, 19.3 ± 0.14 Å and 22.4 ± 0.13 Å in TLR7 (apo), TLR7-R, TLR7-H, TLR7-G systems, respectively. The average distances during the last 20 ns are 11.3 ± 0.08 Å, 10.1 ± 0.08 Å, 14.9 ± 0.12 Å and 11.8 ± 0.05 Å in TLR8 (apo), TLR8-R, TLR8-H, TLR8-G systems, respectively.

chosen as anchor points because they are located on the central axis and closest to the bottom of the two chains. In complex systems, the distance between the two R784 gradually increased in TLR7-R and TLR7-G systems, though the distance in TLR7-G is longer than TLR7-R (Fig 3C). The distance differences between the structures during the last 20 ns MD simulations and initial structure are 7.4 ± 0.15 Å and 7.1 ± 0.13 Å in TLR7-R and TLR7-G systems, respectively. However, the distance changes in TLR7-H, TLR8-R, TLR8-H and TLR8-G systems are not evident (Fig 3C and 3D), which are 2.8 ± 0.14 Å, 1.8 ± 0.08 Å, 3.0 ± 0.12 Å and 2.3 ± 0.05 Å, respectively. In apo systems, the distance changes in TLR7 and TLR8 systems are also not evident (Fig 3C and 3D), which are 3.6 ± 0.15 Å and 0.6 ± 0.08 Å, respectively.

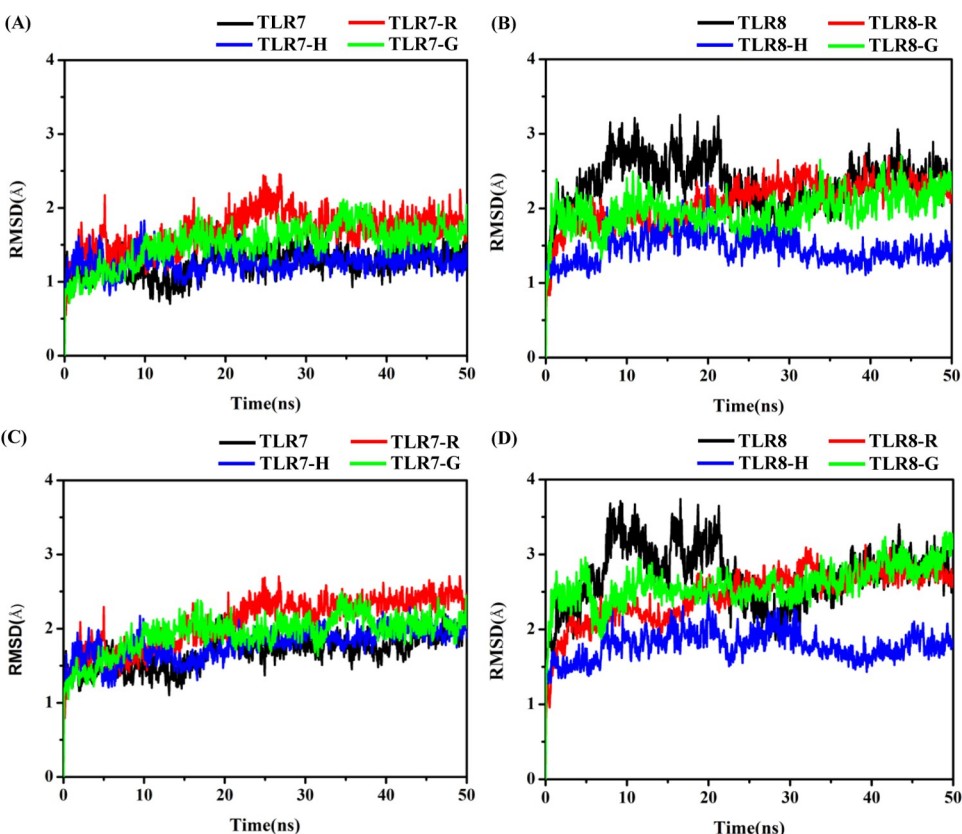

**Fig 4. The RMSD values of the pocket residues in each system defined by 6 Å away from the agonists.** (A) RMSD of backbone atoms of TLR7 pocket. (B) RMSD of backbone atoms of TLR8 pocket. (C) RMSD of heavy atoms of TLR7 pocket. (D) RMSD of heavy atoms of TLR8 pocket.

## 3.2 Conformational changes of pocket residues and agonists

The amino acid residues within 6 Å of the agonists were defined as the pocket residues in this work. The RMSD values of backbone atoms (Fig 4A and 4B) and heavy atoms (Fig 4C and 4D) of pocket residues were calculated respectively. The RMSD values of backbone atoms in the four TLR7 systems change from the initial value (0 ns) to a scope of 1.0–2.0 Å, and 1.5–2.5 Å for heavy atoms, maintaining in this range after 30 ns. In these four systems, the RMSD values of pockets in the TLR7 (apo) and TLR7-H systems are the lowest and the RMSD values of residues in TLR7-R and TLR7-G systems are nearly the same. It reveals that the pocket of TLR7 is less flexible in the TLR7 (apo) and TLR7-H systems. The RMSD values of backbone atoms and heavy atoms in the four TLR8 systems change from the initial value (0 ns) to a scope of 1.2–2.5 Å and 1.5–3.0 Å, maintaining in this scope after 30 ns. In these four systems, the RMSD value of pocket residues in the TLR8-H system is the lowest of the four, while the RMSD values of pocket residues in the TLR8 (apo), TLR8-R and TLR8-G systems are nearly the same. It reveals that the pocket of TLR8 is less flexible in the TLR8-H system compared to TLR8 (apo), TLR8-R and TLR8-G systems. The above analysis suggests that the pockets residues of TLR8 were more flexible than that of TLR7, though the overall conformation change of TLR8 is smaller than that of TLR7 (Fig 3).

We also examined the conformational changes of agonists. The heavy atoms' RMSD of agonists was calculated in the six TLR-agonist systems (Fig 5B and 5E) with respect to the initial conformation. The RMSD of agonists in TLR7 and TLR8 complex systems increase from the

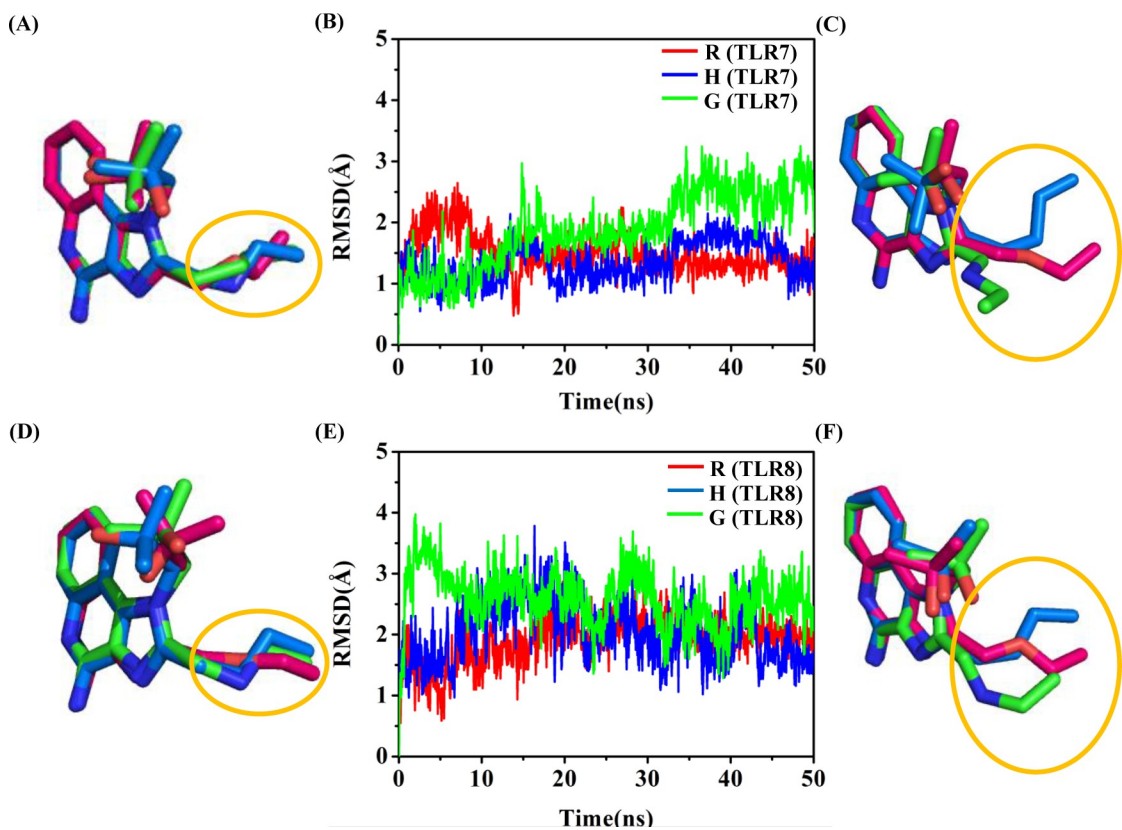

**Fig 5. The conformational change of the agonists.** Schematic diagram of three TLR7 agonists superimposed on the imidazoquinoline ring at 0 ns (A) and 40 ns (C). Schematic diagram of three TLR8 agonists superimposed on the imidazoquinoline ring at 0 ns (D) and 40 ns (F). RMSD values of agonists in TLR7 complex systems (B) and TLR8 complex systems (E).

initial value (0 ns) to a scope of 1.0–3.0 Å and maintain in this scope after 30 ns. However, three agonists show significant differences in fluctuation range. The fluctuations are largest in TLR7-G and TLR8-G systems, and the fluctuations are smallest in TLR7-R and TLR8-R. The conformations of the three agonists were superimposed on the imidazoquinoline ring, the initial frame (0 ns) and a frame near the late stage of the simulation (40 ns) were selected for comparison. In the initial frame, all atoms of the agonists of TLR7 (Fig 5A) and TLR8 (Fig 5D) complex systems are superimposed nicely. The alignment at t = 40 ns of snapshot was chosen to represent every conformation in the simulation 30–50 ns, the side chain orientations of the ligands in TLR7 (Fig 5C) and TLR8 (Fig 5F) are apparently different.

## 3.3 MM-GBSA binding free energy

Aiming to quantify the selectivity profile between the agonists and TLR7 and TLR8, the last 20 ns of simulation data was applied for MM-GBSA binding free energy analysis. The predicted binding free energies of R, H and G binding to TLR7 and TLR8 are summarized in Table 1. The binding free energies of R, H and G binding to TLR7 are -44.21 kcal/mol, -50.50 kcal/mol and -33.65 kcal/mol, respectively. The binding free energies of R, H and G binding to TLR8 are -39.55 kcal/mol, -42.09 kcal/mol and -20.99 kcal/mol. The results indicate that H shows the strongest binding affinity, while G shows the lowest. Additionally, the binding affinity of each of the agonists with TLR7 is stronger than that of TLR8, consistent with previously published experimental data of R, H and G [59, 78–80].

**Table 2. Results of MM-GBSA method of R, H and G to TLR7[b].**

| Energetic contributions | TLR7-H | TLR7-R | TLR7-G |
|---|---|---|---|
| $\Delta E_{vdW}$ | -45.72 ± 0.20 | -45.24 ± 0.23 | -34.60 ± 0.23 |
| $\Delta E_{elec}$ | 398.07 ± 0.84 | 393.98 ± 0.63 | 788.50 ± 1.45 |
| $\Delta E_{polar}$ | -396.95 ± 0.79 | -387.33 ± 0.57 | -783.04 ± 1.35 |
| $\Delta E_{surf}$ | -5.90 ± 0.01 | -5.63 ± 0.01 | -4.52 ± 0.02 |
| $\Delta E_{pol,ele}$ | 1.12 | 6.65 | 5.46 |
| $\Delta G_{binding}$ | -50.50 ± 0.25 | -44.21 ± 0.24 | -33.66 ± 0.31 |

[b]$\Delta E_{pol,ele}$ is $\Delta E_{polar}$ plus $\Delta E_{elec}$. All units are kcal/mol.

To better understand the binding profile, key binding contributors were also analyzed and summarized in Table 2 for TLR7, and in Table 3 for TLR8. Specifically, van der Waals interaction makes a major contribution to binding free energy. In TLR7-H and TLR7-R systems, there are stronger van der Waals interactions compared to the TLR7-G system. In TLR8-H and TLR8-R systems, there are also stronger van der Waals interactions compared to the TLR8-G system. The van der Waals interactions of TLR7 complex systems are overall stronger than that of TLR8 complex systems. The energy is unfavorable to binding is the total of electrostatic interaction and polar solvation energy. TLR7-H and TLR8-H systems have the strongest binding affinity as compared to that of corresponding complex systems, because of lower van der Waals energies and contribution from the electrostatic interaction and polar solvation energy. The above indicates that the pockets of TLR7 and TLR8 are hydrophobic pockets. Furthermore, the energy decomposition values of binding free energy were calculated to get insight into the binding mode of the agonists with TLR7 and TLR8. Items that contributed less than -1.0 kcal/mol to the binding free energy are listed in S1-S6 Tables in S1 File. In S1 and S2 Figs in S1 File, the values of these main residues are displayed in a bar graph.

Fig 6A–6C depicts the interaction plot of H, R and G with TLR7 pocket residues. There are electrostatic interactions, H-bond interactions, π-π interactions, and C-H-π interactions between D555, T586, F408 and L557 with agonists. F351, Y356, V381 and I585 form the hydrophobic interaction regions. Compared with H and R, G forms a stronger H-bond interaction with Thr586 of TLR7 through a nitrogen atom. However, the movement of the side chain weakens the van der Waals interactions between G and F351, V381 and I585. Compared with R, the carbon atom on the side chain of H is more flexible and hydrophobic than the oxygen atom on the side chain, leading H to be more adapted to pocket environment. H forms stronger binding free energies with D555, V381 and F351 (-6.768, -1.450 and -3.503 kcal/mol) than R (-4.542, -0.964 and -2.849 kcal/mol).

**Table 3. Results of MM-GBSA method of R, H and G to TLR8[c].**

| Energetic contributions | TLR8-H | TLR8-R | TLR8-G |
|---|---|---|---|
| $\Delta E_{vdW}$ | -41.41 ± 0.22 | -38.32 ± 0.19 | -27.90 ± 0.19 |
| $\Delta E_{elec}$ | 74.98 ± 0.76 | 27.72 ± 0.68 | 157.83 ± 1.26 |
| $\Delta E_{polar}$ | -70.10 ± 0.76 | -23.78 ± 0.63 | -146.77 ± 1.21 |
| $\Delta E_{surf}$ | -5.56 ± 0.02 | -5.16 ± 0.02 | -4.14 ± 0.02 |
| $\Delta E_{pol,ele}$ | 4.88 | 3.94 | 11.06 |
| $\Delta G_{binding}$ | -42.09 ± 0.28 | -39.55 ± 0.20 | -20.99 ± 0.19 |

[c]$\Delta E_{pol,ele}$ is $\Delta E_{polar}$ plus $\Delta E_{elec}$. All units are kcal/mol.

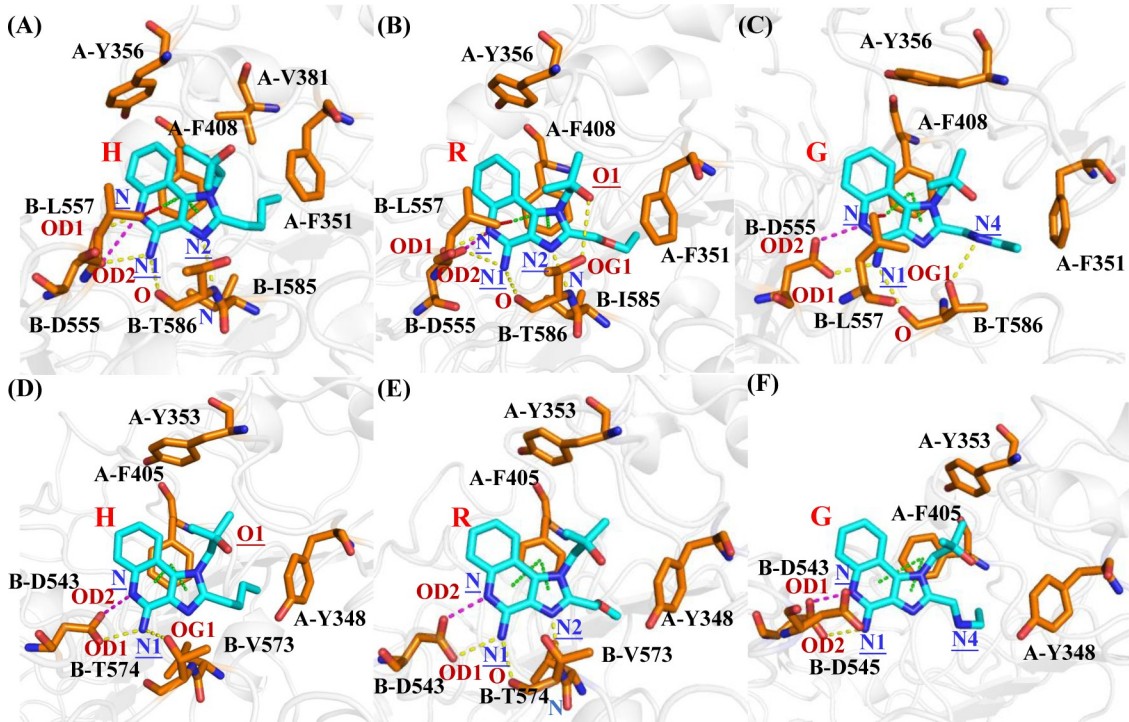

**Fig 6. The interactions between the agonists R, H and G against TLR7 and TLR8.** TLR7-H system (A), TLR7-R system (B), TLR7-G system (C), TLR8-H system (D), TLR8-R system (E), TLR8-G system (F). The electrostatic interactions are shown in magentas dashes, the H-bond interactions are shown in yellow dashes, the π-π interactions are shown in green dashes, and the C-H-π interactions are shown in red dashes. TLR7 and TLR8 are shown in white cartoon. Agonists and representative residues are shown in cyan and orange stick.

Fig 6D–6F depicts the interaction plot of H, R and G with of TLR8 pocket residues. There are electrostatic interactions, H-bond interactions and π-π interactions between D543, T574 and F405 with H and R. Y348, Y353, V378 and V573 form the hydrophobic interaction regions. Compared with H and R, G forms two electrostatic interactions with D545 and D543. However, this interaction weakens the H-bond interaction with T574 and Van der Waals interaction with Y348 and V573. Zhu et al proposed that G is an agonist of TLR7 but not of human TLR8, which in accordance with our modeling. Using NF-κB reporter assay to measure the activation of human TLR7 and human TLR8, they found that G only activated human TLR7, but not human TLR8 in Cos-7 cells and 293T cells [80]. Compared with R, the carbon atom on the side chain of H is more flexible and hydrophobic than the oxygen atom on the side chain, leading H to be more adapted to pocket environment. It forms stronger electrostatic interaction with D543 and stronger van der Waals interactions with Y353.

Agonists R, H and G form C-H-π interaction with L557 of TLR7, which has been also reported in other study for R [55]. However, in TLR8 the corresponding residue of L557 is D545, which could not form strong interactions with agonists due to its shorter side chain.

## 3.4 Hydrogen bond interaction and van der Waals interaction between TLR7/8 and agonists

To further explore the interactions between three agonists and the receptors, the occupancy of hydrogen bonds with more than 10% occupancy between the agonists and residues atoms were analyzed to discard the extremely weak hydrogen bond interaction. The important atoms

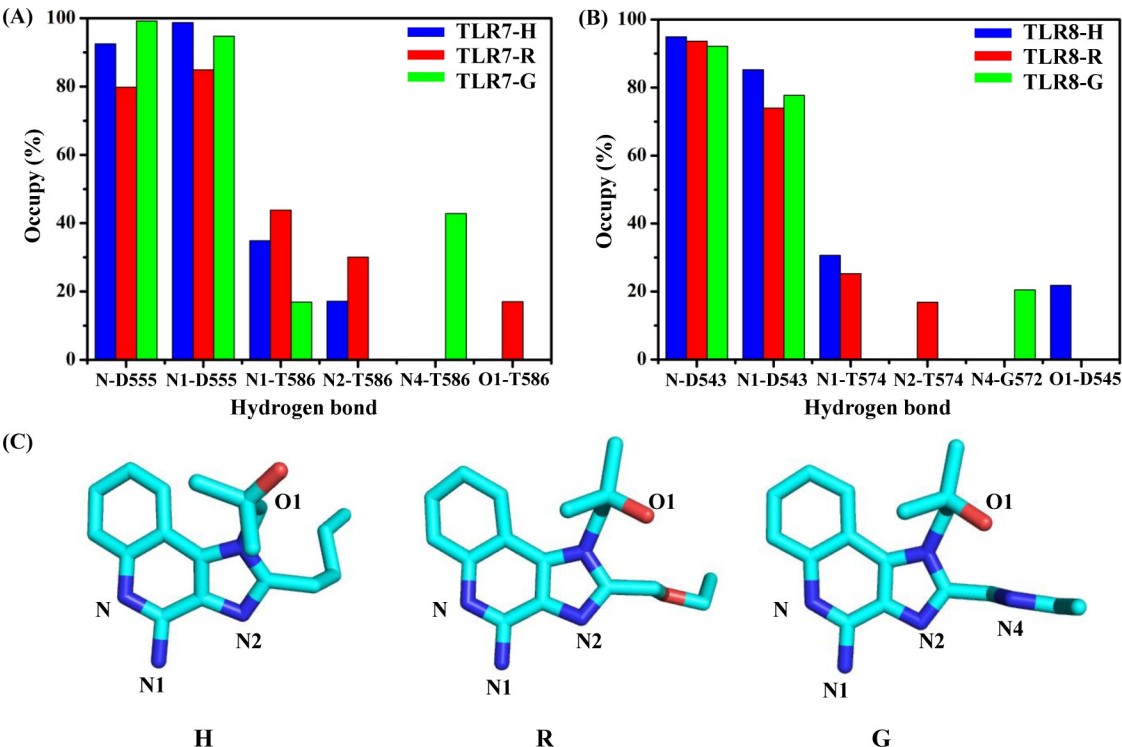

**Fig 7. Analysis of hydrogen bond interactions between three agonists and TLR7 and TLR8.** Occupancy of hydrogen bonds between agonists and TLR7. (A) and TLR8 (B). (C) The position of the important atoms on the agonists H, R and G.

on the three agonists are shown in Fig 7C. The details of the hydrogen bond between agonists and TLR7 are shown in Fig 7A; S7-S9 Tables in S1 File. In TLR7-H, TLR7-R and TLR7-G systems, the stable hydrogen bonds formed between D555 and N and N1 atoms of the agonists were maintained during the last 20 ns of the conformations. In TLR7-H and TLR7-R systems, the occupancy of hydrogen bonds between T586 and N1 atoms of the agonists (TLR7-H: 34.88%, TLR7-R: 43.83%) are similar. Nevertheless, the occupancy of the hydrogen bond between N1 and T586 of TLR7-G was very low (TLR7-G: 12.99%). Since N4 on G was protonated, a hydrogen bond formed between T586 of TLR7-G and N4, resulting in a change in the orientation of the side chain and hence the loss of the hydrogen bond between T586 of TLR7-G and N2.

The details of the hydrogen bonds between three agonists and TLR8 are shown in Fig 7B; S10-S12 in Tables in S1 File. In TLR8-H, TLR8-R and TLR8-G systems, the stable hydrogen bonds formed between D543 and N and N1 atoms of the agonists were maintained during the last 20 ns of the conformations. In TLR8-H and TLR8-R systems, the hydrogen bonds form between T574 of TLR8 and N1 (TLR8-H: 30.68%, TLR8-R: 25.29%). Since N4 on G was protonated, hydrogen bond formed between G572 of TLR8-G and N4, which changed the orientation of the side chain.

The occupancy of residues less than 5 Å away from the agonists were analyzed in detail. To analyze the difference in data, all occupancy less than 20% or close to 100% in all six systems were ignored. The TLR7-R (Fig 8C) and TLR8-R (Fig 8D) systems are chosen to present the position relationship between residues and agonists. As shown in Fig 8A, the occupancy of residues N265, F349, E352, L353, G354, G379, Y380, T406, N407, F466, Y579 and H587 are highest in TLR7-H, lower in TLR7-R, and lowest in TLR7-G. Among them, residues N265, F349, E352, L353, Q379, Y380, T406, N407 and F466 are located around the aliphatic tail. As shown

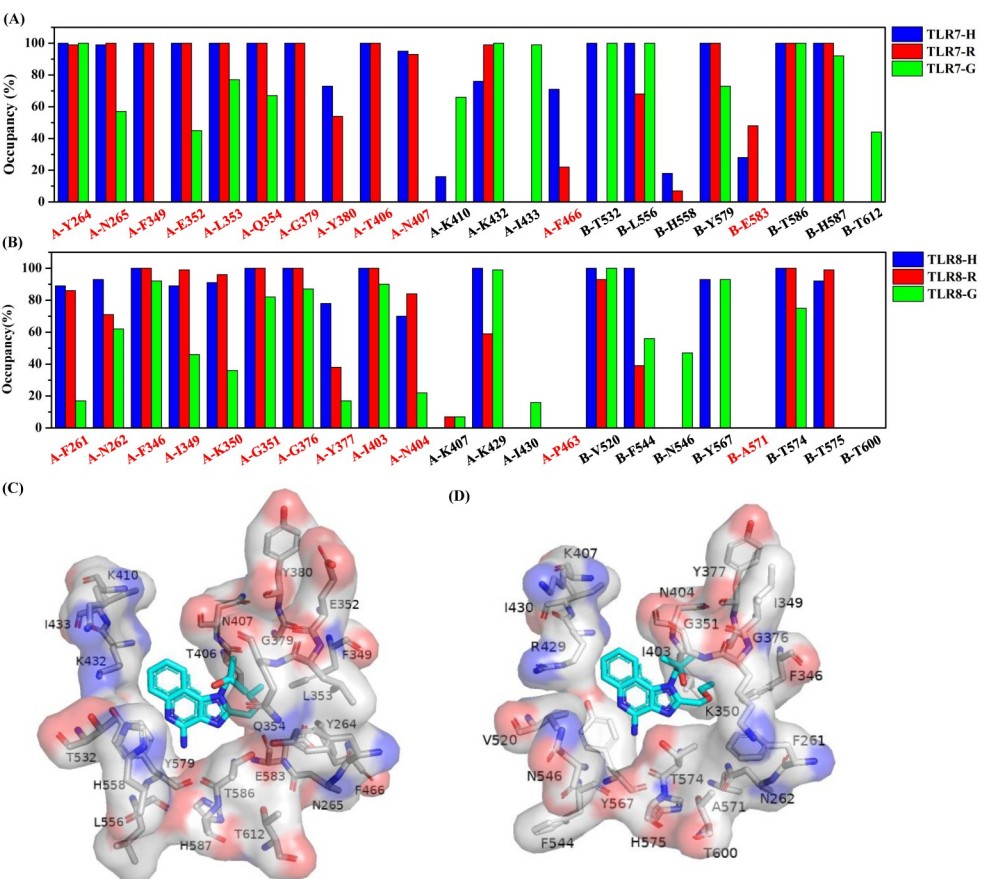

**Fig 8. Occupancy of residues less than 5 Å away from the three agonists.** TLR7 complex systems (A) and TLR8 complex systems (B). Red labels of horizontal coordinate represent pockets residues around side chain of agonists. The residues on x-axis of (A) are in alignment with that of (B). Position relationship between these residues and agonist in TLR7-R system (C) and TLR8-R system (D). The residues are shown in surface and stick. Red surface represents negative charge and blue surface represents positive charge. Agonists are shown in cyan stick.

in Fig 8B, the occupancy of residues F261, N262, F346, G351, G376, Y377, I403 and T574 are highest in TLR8-H, lower in TLR8-R and lowest in TLR8-G. Among them, residues F261, N262, F346, G376, Y377 and I403 are located around the aliphatic tail. The above indicates that the occupancy of residues around three aliphatic tails are quite different, resulting in the van der Waals interaction between three agonists and receptors to be ranked H, R, then G, from highest to lowest. Comparing Fig 8A and 8B, the residues F466 and E583 in TLR7 complex systems were located in the 5 Å range around the agonists for a certain period of time, while the occupancy of corresponding residues P463 and A571 in TLR8 complex system is zero. Further, the occupancy of residues Y264, N265, L353 and H587 in three TLR7 complex systems are higher than residues F261, N262, L350, and H575 in three TLR8 complex systems, respectively. In addition, the occupancy of residues E352, L556 and T586 in two or one TLR7 complex systems is higher than I349, F544 and T574 in TLR8 complex systems. For example, the occupancy of residues E352 in TLR7-H complex system and L556 in TLR7-R and TLR7-G complex systems and T586 in TLR7-G system is higher than I349, F544 and T574 in TLR8 corresponding systems. This explains why the van der Waals interaction and binding affinity between agonists and TLR7 is stronger than TLR8. The corresponding counterpart of the residue of TLR7 in Fig 8A is the residue of TLR8 in Fig 8B.

## Conclusion

The aim of this research was to explore the intrinsic mechanisms underlying the selectivity of R, H and G for TLR7 and TLR8 at atomic level. MD simulations and MM-GBSA method were used to model the overall conformational changes and calculate the binding free energies between three agonists and the receptors TLR7 and TLR8. Trajectory analysis showed that TLR7-R and TLR7-G systems formed open conformations during the simulation, however, other systems kept in closed conformations. The pocket residues in TLR7 are conformationally less flexible than those in TLR8, suggesting tight binding in TLR7. This is confirmed by the predicted binding free energies via MMM-GBSA method. Moreover, the calculated binding free energies indicated that the three agonists are more sensitive for TLR7 than TLR8, and the rank of the binding free energy values are in agreement with the experimental EC50 values in the cellular assay. In brief, in the last 20 ns of the complex systems, the flexible and hydrophobic aliphatic side chain of H forms van der Waals interactions with V381 and F351 of TLR7 and Y353 of TLR8. The side chain nitrogen of G is positively charged in an acidic environment, leading to less favorable interactions with I585 of TLR7 and V573 of TLR8. Stable hydrogen bonds were formed between agonists and D555 of TLR7 and D543 of TLR8. The occupancy of residues around less than 5 Å away from three agonists is quite different, which account for the deviation of van der Waals interaction between agonists and receptors. An atomic difference on the aliphatic tail of each agonist results in the occupancy of residues and the change of van der Waals interaction. Thus, MD simulations provide explanation of differences in interaction modes of three agonists binding with TLR7 and TLR8 at the atomic level, paving the way for further design of more effective TLR7 and TLR8 modulators.

## Supporting information

**S1 File. Energy decomposition of binding free energy of agonists to TLR7 and TLR8.** Details of hydrogen bonds between agonists and TLR7 and TLR8. The binding free energies between each residue of TLR7 and TLR8 and agonists.
(DOCX)

## Author Contributions

**Conceptualization:** Steven Zhang, Jinxia Nancy Deng.

**Data curation:** Xiaoyu Wang.

**Formal analysis:** Xiaoyu Wang, Yu Chen, Jinxia Nancy Deng.

**Funding acquisition:** Jinxia Nancy Deng.

**Investigation:** Xiaoyu Wang, Yu Chen, Jinxia Nancy Deng.

**Methodology:** Xiaoyu Wang.

**Project administration:** Jinxia Nancy Deng.

**Resources:** Jinxia Nancy Deng.

**Supervision:** Jinxia Nancy Deng.

**Validation:** Xiaoyu Wang.

**Visualization:** Xiaoyu Wang.

**Writing – original draft:** Xiaoyu Wang, Yu Chen.

**Writing – review & editing:** Xiaoyu Wang, Yu Chen, Steven Zhang, Jinxia Nancy Deng.

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
