## [Decision Letter · Decision Letter 0]

16 Dec 2021

PONE-D-21-35895Molecular dynamics simulations reveal the selectivity mechanism of structurally similar agonists to TLR7 and TLR8PLOS ONE

Dear Dr. Jinxia Nancy Deng,

Thank you for submitting your manuscript to PLOS ONE. After careful consideration based on the reviewers comments, we feel that it has merit but does not fully meet PLOS ONE’s publication criteria as it currently stands and must undergo a major revision for further consideration. Therefore, we invite you to submit a revised version of the manuscript that addresses the points raised during the review process.

We look forward to receiving your revised manuscript.

Kind regards,

Anand Gaurav

Academic Editor

PLOS ONE

Journal Requirements:

[The authors have declared that no competing interests exist.]

We note that one or more of the authors are employed by a commercial company: Shanghai ChemPartner Co., Ltd., 576 Libing Road, Shanghai. 

Please include both an updated Funding Statement and Competing Interests Statement in your cover letter. We will change the online submission form on your behalf

Reviewers' comments:

Reviewer's Responses to Questions

**Comments to the Author**

1. Is the manuscript technically sound, and do the data support the conclusions?

Reviewer #1: Yes

Reviewer #2: Yes

2. Has the statistical analysis been performed appropriately and rigorously? 

Reviewer #1: Yes

Reviewer #2: Yes

3. Have the authors made all data underlying the findings in their manuscript fully available?

Reviewer #1: Yes

Reviewer #2: Yes

4. Is the manuscript presented in an intelligible fashion and written in standard English?

Reviewer #1: Yes

Reviewer #2: Yes

5. Review Comments to the Author

Reviewer #1: The article entitled "Molecular dynamics simulations reveal the selectivity mechanism of structurally similar agonists to TLR7 and TLR8". Through molecular dynamics simulations, the authors aim to understand the mechanism of selective profiles of small molecule modulators against TLR7 and TLR8. The article falls within the scope of PLOS ONE, and may be considered for publication subject to rectification of the following issues:

1. Introduction is lengthy, kindly reduce the MD simulations part in the introduction section.

2. Page 14, line 295: However, the movement of the side chain weakens the van der Waals interactions between G and Phe351, Val381 and Ile585.

line 298: It forms stronger electrostatic interactions with Asp555 and stronger van der Waals interaction with Val381 and Phe351.

These two statements are contradictory. In the first statement, you mentioned the movement of the side chain weakens the van der Waals interactions between G and Phe351 and Val381. This means van der Waals interactions are weak in this complex.

In the second statement, you mentioned molecule-H of G forms stronger van der Waals interaction with Val381 and Phe351. The van der Waals interactions in TLR7-G complex are weak or strong interactions? Kindly re-write this part.

3. In Fig6C, the residues Val381 and Ile585 do not appear in the figure, kindly include these two residues in the figure if they are important for the interactions as mentioned in page 14, line 296 and 299.

4. In Fig6F, the residues Thr574 and Val573 do not appear in the figure, kindly include these two residues in the figure as you mentioned there are H-bond interaction with Thr574 and Van der Waals interaction with Val573, page 14, line 304 and 305.

5. Page 14, line 302, you have mentioned interactions with Val378 of TLR8 but in Fig6D-E Val378 do not appear in any figure. Kindly include the residue in the figures.

6. There are many errors in English that should be improved. kindly use the same tense in your manuscript.

7. Errors need to be corrected as follow:

1. page 4, line 91: Up to now, MD simulations have been successfully applied to many large systems....million atoms.

Change to “Up to date, MD simulations have been successfully applied to many large systems....million atoms.”

2. Page 5, line 110: In general, they are more potent on TLR7 than those on TLR8 (Table 1).

Change to "In general, they are more potent on TLR7 than TLR8 (Table 1).

3. Page 7, line 160: The binding free energy (ΔGbinding) between a receptor and a ligand and can be estimated using Equation 1:

Change to "The binding free energy (ΔGbinding) between a receptor and a ligand can be estimated using Equation 1:"

4. Page 7, line 166: ΔGbinding: the relative binding free energy. Kindly move this sentence to the beginning of the paragraph (line 163) before ΔEMM : the gas phase...

5. Page 8, line 187: It can be observed that the RMSD value in the TLR8 (apo).....that the TLR8 is more flexible in the presence of agonist. Is it presence or absence, kindly check as higher RMSD is associated with protein flexibility.

6. Page 10, line 239: On the other hand of the trajectory analysis, we also examine the conformational change of agonists.

Change to "On the other hand of the trajectory analysis, we also examine the conformational changes of the agonists."

7. Page 11, line 241: The RMSD of agonists in TLR7 and TLR8 complex systems increase from the initial value (0 ns) to a scope of 1.0-3.0 Å and all reach maintain in this scope after 30 ns.

Change to "The RMSD of the agonists in TLR7 and TLR8 complex systems increase from the initial value (0 ns) to a scope of 1.0-3.0 Å and maintain in this scope after 30 ns.

8. Page 11, line 246: The conformations of the three agonists were superimposed on the imidazoquinoline ring, and the initial frame (0 ns) and a frame near the late stage of the simulation (40 ns) were selected for comparison.

Change to "The conformations of the three agonists were superimposed on the imidazoquinoline ring, the initial frame (0 ns) and a frame....comparison."

9. Page 11, line 248: The alignment at t= 40 ns of snapshot was chose to represent every conformation in the simulation 30-50 ns, the side chain orientations of the ligands in TLR7 (Fig 5C) and this in TLR8 (Fig 5F) are apparently different.

Change to "The alignment at t= 40 ns of snapshot was chosen to represent every conformation in the simulation 30-50 ns, the side chain orientations of the ligands in TLR7 (Fig 5C) and TLR8 (Fig 5F) are apparently different.

10. Page 15, line 324: Agonists and representative residues are shown in orange and cyan stick.

Change to "Agonists and representative residues are shown in cyan and orange stick."

11. Page 15, line 333: In TLR7-H, TLR7-R and TLR7-G systems, the stable hydrogen bonds formed in Asp555 and N and N1 atoms on agonists were maintained during the last 20 ns of the conformations.

Change to "In TLR7-H, TLR7-R and TLR7-G systems, the stable hydrogen bonds formed between Asp555 and N and N1 atoms of the agonists were maintained during the last 20 ns of the conformations."

12. Page 15, line 334: In TLR7-H and TLR7-R systems, the occupancy of hydrogen bonds between Thr586 and N1 atoms on agonists (TLR7-H: 34.88%, TLR7-R: 43.83%) are similar.

Change to "In TLR7-H and TLR7-R systems, the occupancy of hydrogen bonds between Thr586 and N1 atoms of the agonists (TLR7-H: 34.88%, TLR7-R: 43.83%) were similar."

13. Page 15, line 336: Nevertheless, the occupancy of the hydrogen bond between N1 and Thr586 of TLR7-G is very low (TLR7-G: 12.99%).

Change to "Nevertheless, the occupancy of the hydrogen bond between N1 and Thr586 of TLR7-G was very low (TLR7-G: 12.99%)."

14. Page 15, line 338: Since N4 on G was protonated, a hydrogen bond formed between Thr586 of TLR7-G and N4, resulting in a change the orientation of the side chain and the loss of the hydrogen bond between Thr586 of TLR7-G and N2.

Change to "Since N4 on G was protonated, a hydrogen bond formed between Thr586 of TLR7-G and N4, resulting in a change in the orientation of the side chain and loss of the hydrogen bond between Thr586 of TLR7-G and N2."

15. Page 15, line 342: The details of the hydrogen bonds... formed in Asp543 and N and N1 atoms on agonists were maintained during the last 20 ns of the conformations.

Change to "The details of the hydrogen bonds... formed between Asp543 and N and N1 atoms of the agonists were maintained during the last 20 ns of the conformations.

16. Page 16, line 344: The occupancy of hydrogen bond between N1 and Thr574 of TLR8-G is zero. Do you mean no hydrogen bond formed between Thr574 and N1 of G in TLR8-G complex?

17. Page 16, line 359: are located around the aliphatic tail. And as shown in Fig 8B, the....TLR8-G.

Change to "are located around the aliphatic tail. As shown in Fig 8B, the....TLR8-G."

18. Page 16, line 367: complex system is zero. And the occupancy...respectively.

Change to "complex system is zero. Further, the occupancy...respectively.

19. Page 17, line 388: MD simulations and MM-GBSA method were used to model the overall conformational changes and calculate the binding free energies between three agonists and the TLR7 and TLR8.

Change to "MD simulations and MM-GBSA method were used to model the overall conformational changes and calculate the binding free energies between three agonists and the receptors TLR7 and TLR8."

20. Page 17, line 389: Trajectory analysis showed that TLR7-R and TLR7-G systems form more “open” conformations during the simulation, however, other systems kept in closed conformations.

Change to "Trajectory analysis showed that TLR7-R and TLR7-G systems formed open conformations during the simulation, however, other systems kept in closed conformations.

21. Page 17, line 392-393: Plus, the calculated binding free energies indicated that three agonists are more sensitive for TLR7 than TLR8, and the rank of the binding free energy values are in agreement with the experimental EC50 values in the cellular assay.

Change to "Moreover, the calculated binding free energies indicated that the three agonists are more sensitive for TLR7 than TLR8, and the rank of the binding free energy values are in agreement with the experimental EC50 values in the cellular assay.

22. Page 18, line 397: The side chain nitrogen of G is positively charged in an acidic environment, leading to its much less favorite interaction with Ile585 of TLR7 and Val573 of TLR8.

Change to "The side chain nitrogen of G is positively charged in an acidic environment, leading to less favorite interaction with Ile585 of TLR7 and Val573 of TLR8.

Reviewer #2: The present work describes routine but useful simulations to understand the selectivity of agonists towards TLR7 and TLR8. The study was guided by binding free energy analysis, hydrogen bond analyses, decomposition of free energy. All the data have been presented either in main or supplementary text. The results obtained were almost aligned with the experimental results and helpful in determining the selectivity mechanisms. However the manuscript still require few explanations and language corrections.

Suggestions for clarification or modifications:

Line 13: replace “Highly” with “high”

Line 25: Replace “experiment” with “experimental”

Line 28: Replace “favor” with “favorable”

Line 29: Replace “agonist in” with “agonist of”

Line 29: Replace “imidazoquinolines” with “imidazoquinoline derivatives”  

Line 31: Replace “selective” with “selectivity”

Line 32: Insights for “designing” more potent

Line 36: Molecular dynamics simulations

Line 50: both are located

Line 57: diseases and can be used as good adjutants of vaccines

Line 62: “many distributional and functional”

Line 71: “selectivity mechanism of small molecule agonists of TLR7 and TLR8d TLR8”

Line 80: pH

Line 89: replace “structural” to “structure”

Line 100: replace “against” with “for”

Line 111: replace “first” with initially

Line 112: remove “respectively”. Replace second with “thereafter”

Line 116: replace “selectivity mechanism” with “selectivity”

Line 128-29: The TLR7-H system was modeled based on coordinates of TLR7-R by replacing the oxygen to carbon atom. (Mention the aliphatic chain)

Section 2.2  change heading to “Molecular dynamics simulations”

Line 148: replace Cl^-1^ to Cl^-^

Line 149: “steepest descent algorithm”

Line 154: 50ns is not a long-time simulation. Just mention 50 ns simulation.

Section 2.3 equations must be included for “∆EMM : the gas phase energy consisting of electrostatic (∆Eelec) and van der Waals (∆EvdW) terms. ∆Gsolv : solvation energy including both the polar solvation energy, ∆Gpolar and the nonpolar solvation component, ∆Gsurf”

Line 192: “confirmational”

Line 197: capitalize “_In_”

Line 201:  remove “are both”.

Line 202: “The distance changes in TLR7-H, TLR8-R, TLR8-H and TLR8-G systems are not evident (Fig 3C and Fig 3D). In apo systems, the distance changes in TLR7 and TLR8 systems are also not evident (Fig 3C and Fig 3D).”

This point should be made clearer by showing the distance (angstrom) in Fig 3A and 3B. or by superimposing TLR7 and TLR8 with highlighted anchor residue distances.

Line 217: replace scope with “range”

Line 235: replace “On the other hand of the” by “In another”

& change to “changes”

Line 245: replace chose to “chosen”

Line 246: remove “this is”

Line 256: TLR7 and TLR8

Line 260: while G shows the lowest

Line 270: energy is unfavorable to binding include the values here.

Table 1. Comparison of predicted binding free energies and experimental EC50 values of R, H and G with TLR7 and TLR8

SEM (standard error of means) should be reported for all the BFE values e.g -44.21 ± ?? in all the tables.

Table 2. replace “Item” to “Energetic contributions”  All energetic contributions should have uniform representation e.g. ∆Evdw, ∆Eele……

All units are in kcal/mol.

Line 293: “It forms stronger electrostatic interactions with Asp555 and stronger van der Waals interaction with Val381 and Phe351” should be supported with energy values

Line 306-09: these are repeated lines. Remove.

Line 310: “The residue of TLR8 alignment to Leu557 of TLR7 is Asp545, which might result in a weaker interaction with agonists.” rewrite to make it more understandable.

Line 327: Similar codes (either 3 letter or single) for amino acids should be used in text and figures.

Line 332: side chain and hence the

Caption Fig 7.  “the important atoms”

Line 364: two or one TLR7 please specify

Line 382: remove “the” before TLR7 and TLR8.

Line 384: “pockets” to pocket

Line 392: less favorable interactions

Line 396: each agonist

Line 394: quiet to “quite”

Need to address these questions?

How have you treated long range electrostatic?Any specific reason for using only MM-GBSA (Not MM-PBSA) protocol for binding free energy calculations.**********

6. PLOS authors have the option to publish the peer review history of their article (what does this mean?). If published, this will include your full peer review and any attached files.

Reviewer #1: No

Reviewer #2: No

---

## [Author Response · Author response to Decision Letter 0]

10 Feb 2022

Responses to reviewer#1

The article entitled "Molecular dynamics simulations reveal the selectivity mechanism of structurally similar agonists to TLR7 and TLR8". Through molecular dynamics simulations, the authors aim to understand the mechanism of selective profiles of small molecule modulators against TLR7 and TLR8. The article falls within the scope of PLOS ONE, and may be considered for publication subject to rectification of the following issues:

We greatly appreciate the reviewer’s valuable and constructive suggestions. We have incorporated all the suggestions of the reviewer that helped to improve this manuscript, and highlighted the revisions in red. We hope that you will find the revised manuscript suitable for publication in PLOS ONE. 

1. Introduction is lengthy, kindly reduce the MD simulations part in the introduction section.

Response: Many thanks for the constructive suggestion. The MD simulations part has been reduced and revised paragraph in section 1 is following:

Molecular dynamics (MD) simulation is a very established computational technique to understand the protein structure-function relationship and guide the drug design. In the late 70s, the first case of MD is Bovine pancreatic trypsin inhibitor [26]. Especially in recent years, benefit from the development of modern hardware and force fields, MD simulation has made huge progress. Protein conformation dynamics that is hardly available by current experimental techniques can be observed using MD simulations [27-34]. MD simulation also has played an important role in characterizing receptor-ligand interaction [35-37], providing the guidance of structure-based drug design [38-40] and typical conformations for virtual screening [41-46]. Besides, it helps to reveal the novel binding sites which have not been captured by NMR and X-ray crystallographic analysis, for example, cryptic binding sites in HIV-1 integrase [47-50]. Up to date, MD simulations have been successfully applied to many large systems, such as the complete HIV1 capsid with 64 million atoms up to 100ns [51-54]. MD simulation has been also instrumental on understanding the protein folding and function regulation with a simulation time of 10-100us [55-57].

2. Page 14, line 295: However, the movement of the side chain weakens the van der Waals interactions between G and Phe351, Val381 and Ile585.

line 298: It forms stronger electrostatic interactions with Asp555 and stronger van der Waals interaction with Val381 and Phe351.

These two statements are contradictory. In the first statement, you mentioned the movement of the side chain weakens the van der Waals interactions between G and Phe351 and Val381. This means van der Waals interactions are weak in this complex.

In the second statement, you mentioned molecule-H of G forms stronger van der Waals interaction with Val381 and Phe351. The van der Waals interactions in TLR7-G complex are weak or strong interactions? Kindly re-write this part.

Response: We are thankful to the reviewer’s suggestions. The sentence “However, the movement of the side chain weakens the van der Waals interactions between G and Phe351, Val381 and Ile585.” describes the molecule G. However, the sentence “It forms stronger electrostatic interactions with Asp555 and stronger van der Waals interaction with Val381 and Phe351.” is following the sentence “The carbon atom on the side chain is more flexible and hydrophobic than the oxygen atom on the side chain, leading the molecule-H to be more adapted to pocket environment.”. Thus, the latter sentence describes the molecule H. To make this discussion clear, we revised these sentences as following:

However, the movement of the side chain weakens the van der Waals interactions between G and F351, V381 and I585. Compared with R, the carbon atom on the side chain of H is more flexible and hydrophobic than the oxygen atom on the side chain, leading H to be more adapted to pocket environment. H forms stronger electrostatic interactions with D555 and stronger van der Waals interaction with V381 and F351.

3. In Fig6C, the residues Val381 and Ile585 do not appear in the figure, kindly include these two residues in the figure if they are important for the interactions as mentioned in page 14, line 296 and 299.

Response: We are thankful to the reviewer’s suggestions. Figure 6 marks the important residues that have the binding energy of less than -1kcal/mol with the agonists. However, the movement of the side chain weakens the van der Waals interactions between G and Phe351, Val381 and Ile585. The binding energy between Val381 and Ile585 and G is more than -1kcal/mol (Val381: -0.190 kcal/mol, Ile585: -0.096 kcal/mol). Thus, residues Val381 and Ile585 are not shown in Figure 6C.

4. In Fig6F, the residues Thr574 and Val573 do not appear in the figure, kindly include these two residues in the figure as you mentioned there are H-bond interaction with Thr574 and Van der Waals interaction with Val573, page 14, line 304 and 305.

Response: Many thanks for the constructive suggestion. Figure 6 marks the important residues that have the binding energy of less than -1kcal/mol with the agonist. However, this interaction weakens the H-bond interaction with Thr574 and Van der Waals interaction with Tyr348 and Val573. The binding energy between Thr574 and Val573 and G is more than -1kcal/mol (Thr574: 0.089 kcal/mol, Val573: -0.211 kcal/mol). Thus, residues Thr574 and Val573 are not shown in Figure 6F.

5. Page 14, line 302, you have mentioned interactions with Val378 of TLR8 but in Fig6D-E Val378 do not appear in any figure. Kindly include the residue in the figures.

Response: Many thanks for the constructive suggestion. Val378 of TLR8 forms good van der Waals interaction with H (-1.372 kcal/mol), R (-1.204 kcal/mol) and G (-1.037kcal/mol). However, the binding free energies between Val378 and H, R and G are -0.790 kcal/mol, -0.911 kcal/mol and -0.622kcal/mol. These values are more than -1 kcal/mol, thus they are not marked in the Figure 6.

6. There are many errors in English that should be improved. Kindly use the same tense in your manuscript. Errors need to be corrected as follow:

Response: We greatly appreciate the reviewer’s valuable suggestions. We have double-checked our manuscript and revised it.

(1) Page 4, line 91: Up to now, MD simulations have been successfully applied to many large systems....million atoms.

Change to “Up to date, MD simulations have been successfully applied to many large systems....million atoms.”

(2) Page 5, line 110: In general, they are more potent on TLR7 than those on TLR8 (Table 1).

Change to "In general, they are more potent on TLR7 than TLR8 (Table 1).

(3) Page 7, line 160: The binding free energy (ΔGbinding) between a receptor and a ligand and can be estimated using Equation 1:

Change to "The binding free energy (ΔGbinding) between a receptor and a ligand can be estimated using Equation 1:"

(4) Page 7, line 166: ΔGbinding: the relative binding free energy. Kindly move this sentence to the beginning of the paragraph (line 163) before ΔEMM : the gas phase...

(5) Page 8, line 187: It can be observed that the RMSD value in the TLR8 (apo).....that the TLR8 is more flexible in the presence of agonist. Is it presence or absence, kindly check as higher RMSD is associated with protein flexibility.

(6) Page 10, line 239: On the other hand of the trajectory analysis, we also examine the conformational change of agonists.

Change to "On the other hand of the trajectory analysis, we also examine the conformational changes of the agonists."

(7) Page 11, line 241: The RMSD of agonists in TLR7 and TLR8 complex systems increase from the initial value (0 ns) to a scope of 1.0-3.0 Å and all reach maintain in this scope after 30 ns.

Change to "The RMSD of the agonists in TLR7 and TLR8 complex systems increase from the initial value (0 ns) to a scope of 1.0-3.0 Å and maintain in this scope after 30 ns.

(8) Page 11, line 246: The conformations of the three agonists were superimposed on the imidazoquinoline ring, and the initial frame (0 ns) and a frame near the late stage of the simulation (40 ns) were selected for comparison.

Change to "The conformations of the three agonists were superimposed on the imidazoquinoline ring, the initial frame (0 ns) and a frame....comparison."

(9) Page 11, line 248: The alignment at t= 40 ns of snapshot was chose to represent every conformation in the simulation 30-50 ns, the side chain orientations of the ligands in TLR7 (Fig 5C) and this in TLR8 (Fig 5F) are apparently different.

Change to "The alignment at t= 40 ns of snapshot was chosen to represent every conformation in the simulation 30-50 ns, the side chain orientations of the ligands in TLR7 (Fig 5C) and TLR8 (Fig 5F) are apparently different.

(10) Page 15, line 324: Agonists and representative residues are shown in orange and cyan stick.

Change to "Agonists and representative residues are shown in cyan and orange stick."

(11) Page 15, line 333: In TLR7-H, TLR7-R and TLR7-G systems, the stable hydrogen bonds formed in Asp555 and N and N1 atoms on agonists were maintained during the last 20 ns of the conformations.

Change to "In TLR7-H, TLR7-R and TLR7-G systems, the stable hydrogen bonds formed between Asp555 and N and N1 atoms of the agonists were maintained during the last 20 ns of the conformations."

(12) Page 15, line 334: In TLR7-H and TLR7-R systems, the occupancy of hydrogen bonds between Thr586 and N1 atoms on agonists (TLR7-H: 34.88%, TLR7-R: 43.83%) are similar.

Change to "In TLR7-H and TLR7-R systems, the occupancy of hydrogen bonds between Thr586 and N1 atoms of the agonists (TLR7-H: 34.88%, TLR7-R: 43.83%) were similar."

(13) Page 15, line 336: Nevertheless, the occupancy of the hydrogen bond between N1 and Thr586 of TLR7-G is very low (TLR7-G: 12.99%).

Change to "Nevertheless, the occupancy of the hydrogen bond between N1 and Thr586 of TLR7-G was very low (TLR7-G: 12.99%)."

(14) Page 15, line 338: Since N4 on G was protonated, a hydrogen bond formed between Thr586 of TLR7-G and N4, resulting in a change the orientation of the side chain and the loss of the hydrogen bond between Thr586 of TLR7-G and N2.

Change to "Since N4 on G was protonated, a hydrogen bond formed between Thr586 of TLR7-G and N4, resulting in a change in the orientation of the side chain and loss of the hydrogen bond between Thr586 of TLR7-G and N2."

(15) Page 15, line 342: The details of the hydrogen bonds... formed in Asp543 and N and N1 atoms on agonists were maintained during the last 20 ns of the conformations.

Change to "The details of the hydrogen bonds... formed between Asp543 and N and N1 atoms of the agonists were maintained during the last 20 ns of the conformations.

(16) Page 16, line 344: The occupancy of hydrogen bond between N1 and Thr574 of TLR8-G is zero. Do you mean no hydrogen bond formed between Thr574 and N1 of G in TLR8-G complex?

Response: Yes. There is no hydrogen bond formed between Thr574 and N1 of G in TLR8-G complex system.

(17) Page 16, line 359: are located around the aliphatic tail. And as shown in Fig 8B, the....TLR8-G.

Change to "are located around the aliphatic tail. As shown in Fig 8B, the....TLR8-G."

(18) Page 16, line 367: complex system is zero. And the occupancy...respectively.

Change to "complex system is zero. Further, the occupancy...respectively.

(19) Page 17, line 388: MD simulations and MM-GBSA method were used to model the overall conformational changes and calculate the binding free energies between three agonists and the TLR7 and TLR8.

Change to "MD simulations and MM-GBSA method were used to model the overall conformational changes and calculate the binding free energies between three agonists and the receptors TLR7 and TLR8."

(20) Page 17, line 389: Trajectory analysis showed that TLR7-R and TLR7-G systems form more “open” conformations during the simulation, however, other systems kept in closed conformations.

Change to "Trajectory analysis showed that TLR7-R and TLR7-G systems formed open conformations during the simulation, however, other systems kept in closed conformations.

(21) Page 17, line 392-393: Plus, the calculated binding free energies indicated that three agonists are more sensitive for TLR7 than TLR8, and the rank of the binding free energy values are in agreement with the experimental EC50 values in the cellular assay.

Change to "Moreover, the calculated binding free energies indicated that the three agonists are more sensitive for TLR7 than TLR8, and the rank of the binding free energy values are in agreement with the experimental EC50 values in the cellular assay.

(22) Page 18, line 397: The side chain nitrogen of G is positively charged in an acidic environment, leading to its much less favorite interaction with Ile585 of TLR7 and Val573 of TLR8.

Change to "The side chain nitrogen of G is positively charged in an acidic environment, leading to less favorite interaction with Ile585 of TLR7 and Val573 of TLR8.

Reviewer #2: The present work describes routine but useful simulations to understand the selectivity of agonists towards TLR7 and TLR8. The study was guided by binding free energy analysis, hydrogen bond analyses, decomposition of free energy. All the data have been presented either in main or supplementary text. The results obtained were almost aligned with the experimental results and helpful in determining the selectivity mechanisms. However the manuscript still requires few explanations and language corrections.

We are thankful to the reviewer’s positive and valuable suggestions. We have incorporated all the suggestions of the reviewer that helped to improve the manuscript. The revisions were highlighted in red in the revised manuscript. We hope that you will find the revised manuscript suitable for the publication in PLOS ONE. 

1. Suggestions for clarification or modifications:

Response: We greatly appreciate the reviewer’s valuable suggestions. We have double-checked our manuscript and revised it.

(1) Line 13: replace “Highly” with “high”

(2) Line 25: Replace “experiment” with “experimental”

(3) Line 28: Replace “favor” with “favorable”

(4) Line 29: Replace “agonist in” with “agonist of”

(5) Line 29: Replace “imidazoquinolines” with “imidazoquinoline derivatives” 

(6) Line 31: Replace “selective” with “selectivity”

(7) Line 32: Insights for “designing” more potent

(8) Line 36: Molecular dynamics simulations

(9) Line 50: both are located

(10) Line 57: diseases and can be used as good adjutants of vaccines

(11) Line 62: “many distributional and functional”

(12) Line 71: “selectivity mechanism of small molecule agonists of TLR7 and TLR8d TLR8”

(13) Line 80: pH

(14) Line 89: replace “structural” to “structure”

(15) Line 100: replace “against” with “for”

(16) Line 111: replace “first” with initially

(17) Line 112: remove “respectively”. Replace second with “thereafter”

(18) Line 116: replace “selectivity mechanism” with “selectivity”

(19) Line 128-29: The TLR7-H system was modeled based on coordinates of TLR7-R by replacing the oxygen to carbon atom. (Mention the aliphatic chain)

(20) Section 2.2 change heading to “Molecular dynamics simulations”

(21) Line 148: replace Cl-1 to Cl-

(22) Line 149: “steepest descent algorithm”

(23) Line 154: 50ns is not a long-time simulation. Just mention 50 ns simulation.

(24) Section 2.3 equations must be included for “∆EMM : the gas phase energy consisting of electrostatic (∆Eelec) and van der Waals (∆EvdW) terms. ∆Gsolv : solvation energy including both the polar solvation energy, ∆Gpolar and the nonpolar solvation component, ∆Gsurf”

(25) Line 192: “confirmational”

(26) Line 197: capitalize “In”

(27) Line 201: remove “are both”.

(28) Line 202: “The distance changes in TLR7-H, TLR8-R, TLR8-H and TLR8-G systems are not evident (Fig 3C and Fig 3D). In apo systems, the distance changes in TLR7 and TLR8 systems are also not evident (Fig 3C and Fig 3D).”

This point should be made clearer by showing the distance (angstrom) in Fig 3A and 3B. or by superimposing TLR7 and TLR8 with highlighted anchor residue distances.

Response: We are thankful to the reviewer for this valuable suggestion. We showed the distance in Fig 3 C and Fig 3D. We added the values of initial distance and average distance during the last 20 ns for six complex systems and two apo systems in section 3.1 as following:

The distance differences between the structures during the last 20 ns MD simulations and initial structure are 7.4 ± 0.15 Å and 7.1 ± 0.13 Å in TLR7-R and TLR7-G systems, respectively. However, the distance changes in TLR7-H, TLR8-R, TLR8-H and TLR8-G systems are not evident (Figs 3C and 3D), which are 2.8 ± 0.14 Å, 1.8 ± 0.08 Å, 3.0 ± 0.12 Å and 2.3 ± 0.05 Å, respectively. In apo systems, the distance changes in TLR7 and TLR8 systems are also not evident (Figs 3C and 3D), which are 3.6 ± 0.15 Å and 0.6 ± 0.08 Å, respectively.

Fig 3. The distance changes of chain A and chain B for TLR7 and TLR8. (A) The R784 of each chain is defined as anchor point on TLR7. (B) The P773, counterpart of R784 of TLR7, of each chain is defined as anchor point on TLR8. (C) The time dependent distance between chainA-R784 and chainB-R784. (D) The time dependent distance between chainA-P773 and chainB-P773. The Δ_distance refers to the difference of the chain A and chain B distance between the structures during the last 20 ns MD simulations and initial structure. The initial distances (0 ns) are 15.1 Å, 12.4 Å, 16.5 Å and 15.3 Å in TLR7 (apo), TLR7-R, TLR7-H, TLR7-G systems, respectively. The initial distances (0ns) are 11.9 Å, 8.3 Å, 11.9 Å and 14.1 Å in TLR8 (apo), TLR8-R, TLR8-H, TLR8-G systems, respectively. The average distances during the last 20 ns are 18.7 ± 0.15 Å, 19.8 ± 0.15 Å, 19.3 ± 0.14 Å and 22.4 ± 0.13 Å in TLR7 (apo), TLR7-R, TLR7-H, TLR7-G systems, respectively. The average distances during the last 20 ns are 11.3 ± 0.08 Å, 10.1 ± 0.08 Å, 14.9 ± 0.12 Å and 11.8 ± 0.05 Å in TLR8 (apo), TLR8-R, TLR8-H, TLR8-G systems, respectively.

(29) Line 217: replace scope with “range”

(30) Line 235: replace “On the other hand of the” by “In another” & change to “changes”

(31) Line 245: replace chose to “chosen”

(32) Line 246: remove “this is”

(33) Line 256: TLR7 and TLR8

(34) Line 260: while G shows the lowest

(35) Line 270: energy is unfavorable to binding include the values here.

(36) Table 1. Comparison of predicted binding free energies and experimental EC50 values of R, H and G with TLR7 and TLR8

(37) SEM (standard error of means) should be reported for all the BFE values e.g -44.21 ± ?? in all the tables.

(38) Table 2. replace “Item” to “Energetic contributions” All energetic contributions should have uniform representation e.g. ∆Evdw, ∆Eele……

All units are in kcal/mol.

(39) Line 293: “It forms stronger electrostatic interactions with Asp555 and stronger van der Waals interaction with Val381 and Phe351” should be supported with energy values.

Response: Many thanks for the insightful comments. As suggested, we have added the energy values to consolidate this sentence, which is also copied as following:

Compared with R, the carbon atom on the side chain of H is more flexible and hydrophobic than the oxygen atom on the side chain, leading H to be more adapted to pocket environment. H forms stronger binding free energies with D555, V381 and F351 (-6.768, -1.450 and -3.503 kcal/mol) than R (-4.542, -0.964 and -2.849 kcal/mol).

(40) Line 306-09: these are repeated lines. Remove.

(41) Line 310: “The residue of TLR8 alignment to Leu557 of TLR7 is Asp545, which might result in a weaker interaction with agonists.” rewrite to make it more understandable.

Response: We are thankful to the reviewer for this valuable suggestion. The sentence has been rewritten in section 3.3 as following:

Agonists R, H and G form C-H-π interaction with L557 of TLR7, which has been also reported in other study for R [64]. However, in TLR8 the corresponding residue of L557 is D545, which could not form strong interactions with agonists due to its shorter side chain.

(42) Line 327: Similar codes (either 3 letter or single) for amino acids should be used in text and figures.

(43) Line 332: side chain and hence the

(44) caption Fig 7. “the important atoms”

(45) Line 364: two or one TLR7 please specify

Response: We are thankful to the reviewer for this valuable suggestion. The sentence has been rewritten as following: 

In addition, the occupancy of residues E352, L556 and T586 in two or one TLR7 complex systems is higher than I349, F544 and T574 in TLR8 complex systems. For example, the occupancy of residues E352 in TLR7-H complex system and L556 in TLR7-R and TLR7-G complex systems and T586 in TLR7-G system is higher than I349, F544 and T574 in TLR8 corresponding systems.

(46) Line 382: remove “the” before TLR7 and TLR8.

(47) Line 384: “pockets” to pocket

(48) Line 392: less favorable interactions

(49) Line 396: each agonist

(50) Line 394: quiet to “quite”

2. Need to address these questions?

(1) How have you treated long range electrostatic?

Response: We are thankful to the reviewer for this valuable suggestion. Long-range electrostatic interaction was calculated using PME with the cutoff 9Å. And this sentence is added to the section 2.2 as following:

Long-range electrostatic interaction was calculated using PME with the cutoff 9Å.

(2) Any specific reason for using only MM-GBSA (Not MM-PBSA) protocol for binding free energy calculations.

Response: According to the researches of Hou et al1, the accuracy of MM/GBSA is comparable to MM/PBSA, even in the test of the dataset including 1864 crystal structures MM/GBSA is better than MM/PBSA. However, it is known that MMPBSA is more time expensive than MM/GBSA. Therefore, MM/GBSA was applied to predict the protein-ligand binding free energy in this study.

[1] Sun H, Li Y, Tian S, Xu L, Hou T. Assessing the performance of MM/PBSA and MM/GBSA methods. 4. Accuracies of MM/PBSA and MM/GBSA methodologies evaluated by various simulation protocols using PDBbind data set. Phys Chem Chem Phys. 2014 Aug 21;16(31):16719-29. doi: 10.1039/c4cp01388c.

---

## [Decision Letter · Decision Letter 1]

11 Mar 2022

PONE-D-21-35895R1Molecular dynamics simulations reveal the selectivity mechanism of structurally similar agonists to TLR7 and TLR8PLOS ONE

Dear Dr. Jinxia Nancy Deng,

Thank you for submitting your manuscript to PLOS ONE. After careful consideration, we feel that it has merit but does not fully meet PLOS ONE’s publication criteria as it currently stands. Therefore, we invite you to submit a revised version of the manuscript that addresses the points raised during the review process.

We look forward to receiving your revised manuscript.

Kind regards,

Anand Gaurav

Academic Editor

PLOS ONE

Journal Requirements:

Reviewers' comments:

Reviewer's Responses to Questions

**Comments to the Author**

1. If the authors have adequately addressed your comments raised in a previous round of review and you feel that this manuscript is now acceptable for publication, you may indicate that here to bypass the “Comments to the Author” section, enter your conflict of interest statement in the “Confidential to Editor” section, and submit your "Accept" recommendation.

Reviewer #1: All comments have been addressed

Reviewer #2: All comments have been addressed

2. Is the manuscript technically sound, and do the data support the conclusions?

Reviewer #1: Yes

Reviewer #2: Yes

3. Has the statistical analysis been performed appropriately and rigorously? 

Reviewer #1: N/A

Reviewer #2: Yes

4. Have the authors made all data underlying the findings in their manuscript fully available?

Reviewer #1: Yes

Reviewer #2: Yes

5. Is the manuscript presented in an intelligible fashion and written in standard English?

Reviewer #1: Yes

Reviewer #2: Yes

6. Review Comments to the Author

Reviewer #1: The authors have addressed all the comments, however, there are additional comments that need to be addressed by authors before the acceptance of the manuscript.

- Page 4, line 77. Kindly remove this part from the paragraph, "In the late 70s, the first case of MD is Bovine pancreatic trypsin inhibitor [26]. Especially in recent years, benefit from the development of modern hardware and force fields, MD simulation has made huge progress. Protein conformation dynamics that is hardly available by current experimental techniques can be observed using MD simulations [27-34]." as it is extra and does not add scientific value to your manuscript.

So the updated paragraph will be "Molecular dynamics (MD) simulation is a very established computational technique to understand the protein structure-function relationship and guide the drug design. MD simulation has played an important role in characterizing receptor-ligand interaction....with a simulation time of 10-100us [55-57].

-Page 11, line 246. "In another trajectory analysis, we also examine the conformational changes of agonists. The heavy atom RMSD of agonists was calculated in the six TLR-agonist systems (Figs 5B and 5E) with respect to the initial conformation."

Kindly change it to " We also examine the conformational changes of agonists. The heavy atoms' RMSD of agonists was calculated in the six TLR-agonist systems (Figs 5B and 5E) with respect to the initial conformation."

-Page 16, line 349. "The occupancy of hydrogen bond between N1 and T574 of TLR8-G is zero."

Kindly remove this sentence since there is no hydrogen bond between N1 and T574 of TLR8-G.

Reviewer #2: (No Response)

---

## [Author Response · Author response to Decision Letter 1]

3 Apr 2022

Responses to reviewer#1

The authors have addressed all the comments, however, there are additional comments that need to be addressed by authors before the acceptance of the manuscript.

Many thanks for taking your time to review our responses. We have incorporated all the suggestions of the reviewer that helped to improve this manuscript, and highlighted the revisions in red.

1. Page 4, line 77. Kindly remove this part from the paragraph, "In the late 70s, the first case of MD is Bovine pancreatic trypsin inhibitor [26]. Especially in recent years, benefit from the development of modern hardware and force fields, MD simulation has made huge progress. Protein conformation dynamics that is hardly available by current experimental techniques can be observed using MD simulations [27-34]." As it is extra and does not add scientific value to your manuscript. 

So the updated paragraph will be "Molecular dynamics (MD) simulation is a very established computational technique to understand the protein structure-function relationship and guide the drug design. MD simulation has played an important role in characterizing receptor-ligand interaction .... with a simulation time of 10-100us [55-57].

Response: Many thanks for the constructive suggestion. This paragraph has been reduced. The revised paragraph in section 1 is following:

Molecular dynamics (MD) simulation is a very established computational technique to understand the protein structure-function relationship and guide the drug design. MD simulation also has played an important role in characterizing receptor-ligand interaction [26-28], providing the guidance of structure-based drug design [29-31] and typical conformations for virtual screening [32-37]. Besides, it helps to reveal the novel binding sites which have not been captured by NMR and X-ray crystallographic analysis, for example, cryptic binding sites in HIV-1 integrase [38-41]. Up to date, MD simulations have been successfully applied to many large systems, such as the complete HIV1 capsid with 64 million atoms up to 100ns [42-45]. MD simulation has been also instrumental on understanding the protein folding and function regulation with a simulation time of 10-100us [46-48].

2. Page 11, line 246. "In another trajectory analysis, we also examine the conformational changes of agonists. The heavy atom RMSD of agonists was calculated in the six TLR-agonist systems (Figs 5B and 5E) with respect to the initial conformation."

Kindly change it to " We also examine the conformational changes of agonists. The heavy atoms' RMSD of agonists was calculated in the six TLR-agonist systems (Figs 5B and 5E) with respect to the initial conformation."

Response: We are thankful to the reviewer’s positive suggestion. The revised paragraph in section 3.2 is following:

We also examined the conformational changes of agonists. The heavy atoms' RMSD of agonists was calculated in the six TLR-agonist systems (Figs 5B and 5E) with respect to the initial conformation.

3. Page 16, line 349. "The occupancy of hydrogen bond between N1 and T574 of TLR8-G is zero."

Kindly remove this sentence since there is no hydrogen bond between N1 and T574 of TLR8-G.

Response: We are thankful to the reviewer’s valuable suggestions. This sentence has been removed.

---

## [Editor Report · Decision Letter 2]

12 Apr 2022

Molecular dynamics simulations reveal the selectivity mechanism of structurally similar agonists to TLR7 and TLR8

PONE-D-21-35895R2

Dear Dr. Jinxia Nancy Deng,

We’re pleased to inform you that your manuscript has been judged scientifically suitable for publication and will be formally accepted for publication once it meets all outstanding technical requirements.

Kind regards,

Anand Gaurav

Academic Editor

PLOS ONE

Additional Editor Comments (optional):

Dear Author,

The revised manuscript addresses all the comments of the reviewers and the manuscript can be accepted in the current form.

Regards

Anand Guarav, PhD
---

## [Editor Report · Acceptance letter]

14 Apr 2022

PONE-D-21-35895R2 

Molecular dynamics simulations reveal the selectivity mechanism of structurally similar agonists to TLR7 and TLR8 

Dear Dr. Deng:

I'm pleased to inform you that your manuscript has been deemed suitable for publication in PLOS ONE. Congratulations! Your manuscript is now with our production department. 

Kind regards, 

on behalf of

Dr. Anand Gaurav 

Academic Editor

PLOS ONE